# Optimizing SAT-Nano for Text-Guided 3D Medical Segmentation using Dice Semimetric Loss

Li Zhi[1], Yaqi Wang[2], and Shuai Wang[1]

[1] Hangzhou Dianzi University, Hangzhou, China
[2] Communication University of Zhejiang, Hangzhou, China
shuaiwang.tai@gmail.com

**Abstract.** This paper presents SAT-Nano-JDT, an approach for the CVPR 2025 Text-guided 3D Biomedical Segmentation Challenge, involving fine-tuning the SAT-Nano baseline with JDTLoss (Dice Semimetric Loss). Pre-trained on 10% of challenge data, SAT-Nano was further trained using a composite loss including JDTLoss, aiming to directly optimize Dice scores and enhance segmentation. On the validation coreset, SAT-Nano-JDT showed mixed results: CT semantic DSC improved to 0.644 (vs. 0.643 baseline) and Microscopy instance DSC TP to 0.310 (vs. 0.292). However, MRI and PET performance did not exceed the baseline. This empirical study explores JDTLoss's utility in refining foundation models, noting the challenges in surpassing strong baselines. The code is available at https://github.com/ricoleehduu/SAT-Nano-JDTLoss.git.

**Keywords:** Foundation Models · 3D Biomedical Image Segmentation · Text-guided Segmentation · Dice Semimetric Loss.

## 1 Introduction

3D biomedical image segmentation is vital for quantitative medical image analysis but is challenged by data complexity and laborious annotation [7]. Text-guided segmentation offers an intuitive alternative, using natural language prompts to define targets [13]. The CVPR 2025 Challenge, "Foundation Models for Text-guided 3D Biomedical Image Segmentation," promotes models for robustly segmenting diverse structures in large-scale 3D datasets. Key challenges include generalization across numerous image-mask pairs, interpreting nuanced text prompts, and developing adaptable foundation models.

Foundation models like SAM ([5], SAM2 [8]) spurred medical adaptations (MedSAM [6], MedSAM2 [7]), though often with limited text-prompt capabilities. While interactive methods (SegVol [1], SAM-Med3D [9], VISTA3D [3], nnInteractive [2]) improved user-guided 3D segmentation using non-textual cues, dedicated text-guided models (BioMedParse [12], CAT [4]) also emerged. The Segment Anything in Radiology (SAT) model [13] is a pivotal, knowledge-enhanced universal model for text-prompted segmentation of 497 classes from 72 datasets, forming the challenge baseline. The provided SAT-Nano, a compact version pretrained on 10% of competition data, offers a strong starting point for further refinement through targeted fine-tuning.

Our work investigates enhancing the SAT-Nano baseline via alternative loss formulations for fine-tuning. While SAT's architecture [13] is robust, the loss function critically impacts segmentation accuracy, especially for complex boundaries and class imbalances, and standard losses may not directly optimize metrics like the Dice score. We fine-tuned SAT-Nano by incorporating JDTLoss (Dice Semimetric Loss) [10] into a composite loss function. JDTLoss directly optimizes the Dice score and is theoretically sound for soft labels (performing identically to soft Dice loss with hard labels). We hypothesized this could yield more refined segmentations. The resulting model is termed SAT-Nano-JDT.

Our contributions are: (1) Applying and systematically evaluating JDTLoss within a composite loss for fine-tuning SAT-Nano for text-guided 3D biomedical segmentation. (2) Detailing a fine-tuning methodology on the challenge coreset. (3) Empirically analyzing SAT-Nano-JDT's performance, offering insights into JDTLoss's practical effects on a strong baseline.

## 2    Method

Our methodology, SAT-Nano-JDT, enhances the official SAT-Nano baseline [13] for the CVPR 2025 Text-guided 3D Biomedical Image Segmentation Challenge by fine-tuning it with a composite loss function incorporating JDTLoss (Dice Semimetric Loss) [10]. This section outlines the baseline architecture, prompt handling, our loss integration, and training.

### 2.1    Baseline Model: SAT-Nano Architecture

We build upon the official SAT-Nano model provided by the challenge organizers. SAT (Segment Anything in Radiology) [13] is a knowledge-enhanced universal segmentation model designed to segment a wide array of anatomical structures from 3D medical images across various modalities, guided by text prompts. The SAT-Nano variant used as our starting point is a more compact version of the full SAT model, pre-trained on 10% of the challenge's training data.

The general architecture of SAT, which we retain, consists of three main components:Text Encoder, Vision Backbone and Decoder. A conceptual overview of our fine-tuning pipeline, which leverages the SAT architecture, is shown in Figure 1. We do not modify the core network architecture of the SAT-Nano baseline; our primary intervention lies in the loss function used during the continued training phase.

### 2.2    Segmentation Decoder and JDTLoss Function

The decoder component of SAT is responsible for generating pixel-wise (or voxel-wise in 3D) segmentation masks based on the integrated image features and text prompt embeddings.

**Loss Function**: While the original SAT model likely employs a combination of standard segmentation losses, our primary contribution is the adoption of the

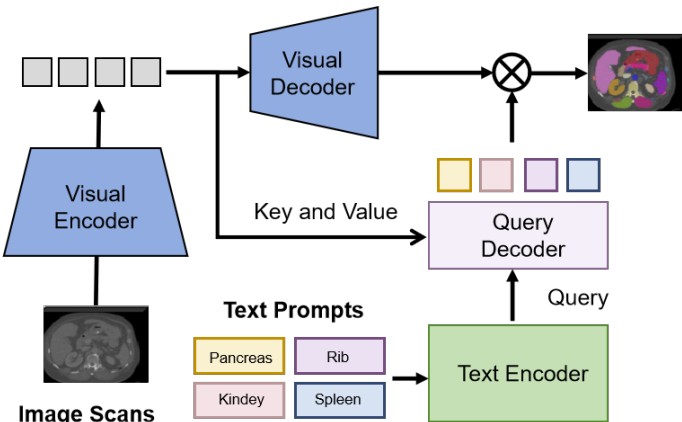

**Fig. 1.** Overview of our fine-tuning pipeline. The pre-trained SAT-Nano model (comprising a text encoder, a 3D U-Net vision backbone, and a segmentation decoder) is further trained on the challenge dataset using the JDTLoss. Text prompts guide the segmentation of target anatomical structures within 3D biomedical images.

JDTLoss (Dice Semimetric Loss) [10] for fine-tuning the SAT-Nano baseline. The Dice Score is a critical metric for evaluating segmentation performance, and loss functions that directly optimize it are often preferred. The standard Soft Dice Loss (SDL) is defined for a prediction $\tilde{x} \in [0,1]^p$ and a hard binary ground truth $y \in \{0,1\}^p$ as:

$$\overline{\Delta}_{\mathrm{SDL},L^1}(\tilde{x},y) = 1 - \frac{2\langle \tilde{x}, y\rangle}{\|\tilde{x}\|_1 + \|y\|_1}$$

A limitation of SDL and similar $L^1$-based losses is their incompatibility with soft labels (where $y \in [0,1]^p$). Although our ground truth labels are hard, the JDTLoss formulation offers a robust alternative that is theoretically sound for soft labels and, importantly, identical to SDL when hard labels are used. This provides a well-grounded optimization objective. We specifically employ one of the Dice Semimetric Losses (DMLs) proposed by Wang et al. [10]. For instance, $\overline{\Delta}_{\mathrm{DML2}}$ is defined as:

$$\overline{\Delta}_{\mathrm{DML2}}(\tilde{x},y) = 1 - \frac{2\langle \tilde{x}, y\rangle}{2\langle \tilde{x}, y\rangle + \|\tilde{x} - y\|_1}$$

where $\tilde{x}$ represents the voxel-wise probabilities from the model's output, $y$ is the ground truth segmentation mask, $\langle \cdot, \cdot \rangle$ denotes the dot product (sum of element-wise products), and $\| \cdot \|_1$ is the $L^1$ norm. According to [10] (Theorem 2), for hard labels $y \in \{0,1\}^p$, $\overline{\Delta}_{\mathrm{DML1}} = \overline{\Delta}_{\mathrm{DML2}} = \overline{\Delta}_{\mathrm{SDL},L^1}$. Thus, by using JDTLoss, we are effectively optimizing the Soft Dice Loss, which is directly related to the Dice score, a primary evaluation metric for this challenge. We hypothesize that this direct optimization, leveraging the well-behaved properties

of DMLs (semimetrics, satisfying reflexivity and positivity), leads to improved segmentation accuracy. In our implementation, JDTLoss replaces or augments the default segmentation loss of the SAT baseline for the final segmentation head.

**Handling Large 3D Inputs**: Medical images, especially 3D volumes, can be very large, exceeding typical GPU memory capacities. The SAT framework, and our fine-tuning process, addresses this using a patch-based approach. During training, crops of size $256 \times 256 \times 96$ are extracted from the full-resolution images. The network then processes these crops, which are further divided into patches of size $32 \times 32 \times 32$ internally by the U-Net architecture if it employs a patch-based mechanism or if this refers to the receptive field granularity. This strategy allows for efficient processing of large volumes while maintaining a manageable memory footprint. During inference, a similar strategy (e.g., sliding window with overlap) is typically used to generate segmentations for the entire volume, which are then aggregated.

### 2.3   Coreset selection strategy

We adopted the official Coreset training dataset for the whole training process.

### 2.4   Post-processing

No specific model-output post-processing steps.

## 3   Experiments

### 3.1   Dataset and evaluation metrics

The development set for the CVPR 2025 Foundation Models for Text-guided 3D Biomedical Image Segmentation Challenge encompasses a broader range of 3D cases sourced from numerous public datasets[3], covering commonly used 3D modalities such as Computed Tomography (CT), Magnetic Resonance Imaging (MRI), Positron Emission Tomography (PET), Ultrasound, and Microscopy images. Our work utilizes the provided 10% coreset of the total training cases for model fine-tuning.

The text-guided segmentation task, which is the focus of our work, includes both semantic segmentation and instance segmentation aspects. For the semantic segmentation task, the primary evaluation metrics are the Dice Similarity Coefficient (DSC) and Normalized Surface Distance (NSD). These metrics assess the segmentation region overlap and boundary distance, respectively, providing a comprehensive measure of segmentation quality. For the instance segmentation task, the evaluation involves computing the F1 score at an overlapping threshold of 0.5, alongside DSC scores for true positive instances. A critical constraint for the challenge is the algorithm runtime, which is limited to 60 seconds per class during inference. Exceeding this time limit results in all DSC and NSD metrics for that specific test case being set to 0.

---

[3] A complete list is available at https://medsam-datasetlist.github.io/

### 3.2   Implementation details

**Preprocessing**  No additional preprocessing strategies were introduced beyond those inherent in the SAT baseline pipeline. For handling large-scale datasets, the SAT framework employs a patch-based approach during training, where crops of size $256 \times 256 \times 96$ are extracted from the images. Data loading is facilitated by 16 worker processes to ensure efficient batch preparation.

**Environment settings**  The development and training of our model were conducted in the environment detailed in Table 1.

**Table 1.** Development environments and requirements.

| Component | Specification |
| --- | --- |
| System | Ubuntu 22.04 LTS |
| CPU | 20 vCPU Intel(R) Xeon(R) Platinum 8457C |
| RAM | 200GB |
| GPU (number and type) | $1 \times$ NVIDIA H20 (96GB HBM, NVLink) |
| CUDA version | 12.4 |
| Programming language | Python 3.12 |
| Deep learning framework | PyTorch 2.5.1, Torchvision |
| Storage | System Disk: 30GB; Data Disk: 50GB |

**Training protocols**  Our fine-tuning procedure commenced from the official SAT-Nano baseline model pre-trained on 10% of the challenge data. The specifics of our training protocol are summarized in Table 2. We utilized nnU-Net style data augmentations as configured in the baseline training script. These augmentations typically include random rotations, scaling, elastic deformations, gamma correction, and mirroring, enhancing the model's robustness and ability to generalize to unseen data.

Our core modification to the training protocol was the integration of the JDTLoss. The overall loss function is a weighted sum of Dice loss, Binary Cross-Entropy (BCE) loss, and JDTLoss, applied at each output layer of the deeply supervised U-Net decoder.

## 4   Results and discussion

### 4.1   Quantitative results on validation set

We evaluated SAT-Nano-JDT, our JDTLoss-enhanced SAT-Nano model, on the CVPR 2025 Challenge coreset-data track. Table 3 benchmarks our approach against the official SAT and CAT baselines.

**Table 2.** Training protocols. (This table summarizes key parameters used for fine-tuning the SAT-Nano baseline with JDTLoss.)

| Parameter | Details |
|---|---|
| Pre-trained Model | SAT-Nano (official baseline, pre-trained on 10% challenge data) |
| Batch size (3D) | 4 |
| Patch size (Input Crop) | $256 \times 256 \times 96$ |
| Internal Patch Size (U-Net) | $32 \times 32 \times 32$ |
| Total steps | Stage 1: 260,000; Stage 2: 100,000 |
| Optimizer | Adam (implied by typical deep learning practices, not explicitly stated in script) |
| Initial learning rate (lr) | Stage 1: $1 \times 10^{-4}$; Stage 2: $1 \times 10^{-5}$ |
| Lr decay schedule | Two-stage schedule with warmup (10,000 steps per stage) |
| Training time (approx.) | [To be filled based on actual training duration, e.g., XX hours on 1xH20] |
| Loss function | Sum of Dice Loss, BCEWithLogitsLoss, and JDTLoss (DML variant), applied with deep supervision. Each component (Dice, BCE, JDT) is equally weighted per deep supervision layer. |
| Number of model parameters | SAT-Nano: 110M (as per SAT paper [13]) |

**Table 3.** Quantitative evaluation on the validation set (coreset-data track). SAT-Nano-JDT refers to our fine-tuned SAT-Nano with JDTLoss.

| Modality | Method | Semantic Segmentation | | Instance Segmentation | |
|---|---|---|---|---|---|
| | | DSC | NSD | F1 | DSC TP |
| CT | CAT Baseline | 0.604 | 0.598 | 0.287 | 0.328 |
| | SAT Baseline | 0.643 | 0.638 | 0.132 | 0.145 |
| | SAT-Nano-JDT | 0.644 | 0.638 | 0.126 | 0.141 |
| MRI | CAT Baseline | 0.426 | 0.488 | 0.185 | 0.283 |
| | SAT Baseline | 0.453 | 0.528 | 0.059 | 0.067 |
| | SAT-Nano-JDT | 0.448 | 0.516 | 0.056 | 0.066 |
| Microscopy | CAT Baseline | - | - | 0.023 | 0.554 |
| | SAT Baseline | - | - | 0.087 | 0.292 |
| | SAT-Nano-JDT | - | - | 0.087 | 0.310 |
| PET | CAT Baseline | - | - | 0.017 | 0.168 |
| | SAT Baseline | - | - | 0.113 | 0.234 |
| | SAT-Nano-JDT | - | - | 0.012 | 0.029 |
| Ultrasound | CAT Baseline | 0.818 | 0.812 | - | - |
| | SAT Baseline | 0.755 | 0.719 | - | - |
| | SAT-Nano-JDT | 0.712 | 0.733 | - | - |

SAT-Nano-JDT demonstrated a marginal improvement in CT semantic DSC (0.644) and microscopy instance DSC TP (0.310) over the SAT baseline. However, the performance in semantic segmentation of MRI (DSC 0.448 vs. 0.453) and segmentation of PET instances (F1 0.012 vs. 0.113) was lower. For Ultrasound, SAT-Nano-JDT showed improved NSD (0.733 vs. 0.719) but lower DSC (0.712 vs. 0.755). These mixed results indicate that while JDTLoss aims to directly optimize Dice-related metrics, its integration into the SAT-Nano finetuning pipeline did not yield uniform enhancements across all modalities and tasks compared to the already strong baseline.

### 4.2 Qualitative results on validation set

Visual assessment of the performance of SAT-Nano-JDT is provided through examples from the validation set, comparing the predictions with ground truth[4]. Fig. 2 will show representative good and challenging segmentations for each modality (CT, MRI, Microscopy, PET, Ultrasound).

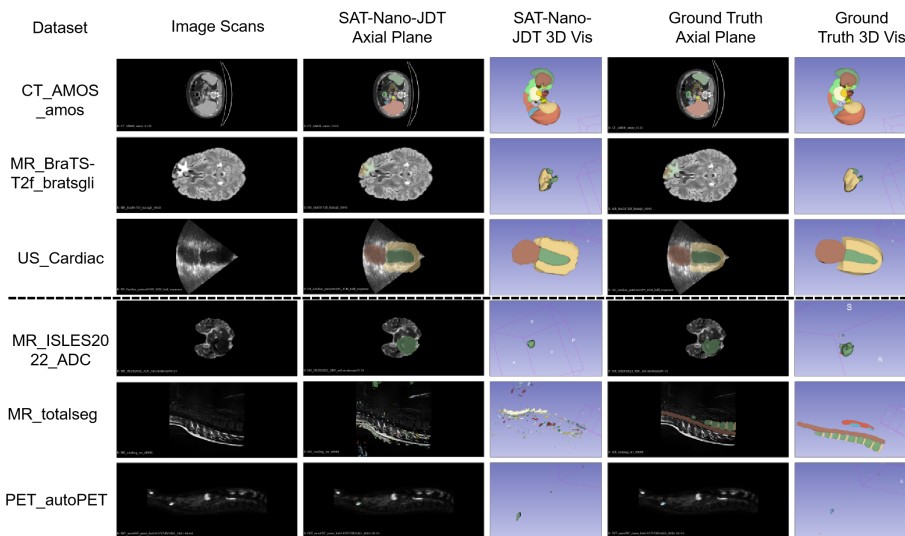

**Fig. 2.** Qualitative CT segmentation by SAT-Nano-JDT. Top row(s) display successful segmentations; bottom row(s) illustrate challenging scenarios.

**Discussion of Failed Cases**: Suboptimal segmentations typically arise from: (i) inherent ambiguity in anatomical boundaries or challenging image contrast, particularly in [mention a specific example if you have one, for example, low contrast MRI lesions]; (ii) potential limitations in generalizing from the 10%

---

[4] Ground truth available at https://huggingface.co/datasets/junma/CVPR-BiomedSegFM/tree/main/3D_val_gt

coreset to diverse validation cases, especially for underrepresented structures or imaging variations like those observed in PET; (iii) difficulties in accurately distinguishing closely located instances solely from text prompts. The performance drop in PET suggests a particular sensitivity, possibly due to the unique characteristics of tracer uptake or the specificity required by text prompts for PET targets not fully captured during fine-tuning with JDTLoss on the coreset.

### 4.3   Results on final testing set

This is a placeholder. No need to show testing results now. We will announce the testing results during CVPR (6.11) then you can add them during the revision phase.

### 4.4   Limitation and Future Work

Our study highlights the nuanced impact of integrating JDTLoss into a pre-trained foundation model. **Limitations** include: (1) The JDTLoss did not uniformly outperform the SAT baseline, suggesting the need for more extensive tuning or that the baseline's original loss configuration is highly optimized. (2) Our exploration was confined to the 10% coreset and a specific JDTLoss integration strategy.

    **Future Work** will focus on: (1) Comprehensive hyperparameter optimization for the JDTLoss component within the composite loss. (2) Evaluation of the approach on the complete training data set to assess scalability and broader impact.

## 5   Conclusion

We presented SAT-Nano-JDT, an adaptation of the SAT-Nano baseline fine-tuned with JDTLoss for text-guided 3D biomedical image segmentation. On the CVPR 2025 Challenge coreset validation set, SAT-Nano-JDT yielded incremental improvements in specific areas like CT semantic DSC and Microscopy instance DSC TP, but did not consistently surpass the strong SAT baseline across all modalities and metrics. This underscores the complexity of enhancing highly optimized foundation models through isolated loss function modifications without extensive re-tuning. Our findings contribute to the understanding of the application of specialized loss functions, indicating that, while theoretically beneficial, their practical efficacy requires careful calibration within the broader training paradigm of large-scale models.

**Acknowledgements** We thank all the data owners for making the medical images publicly available and CodaLab [11] for hosting the challenge platform.

**Disclosure of Interests.** The authors have no competing interests to declare that are relevant to the content of this article.

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

**Table 4.** Checklist Table. Please fill out this checklist table in the answer column. (**Delete this Table in the camera-ready submission**)

| Requirements | Answer |
| --- | --- |
| A meaningful title | Yes |
| The number of authors ($\leq 6$) | 3 |
| Author affiliations and ORCID | Yes |
| Corresponding author email is presented | Yes |
| Validation scores are presented in the abstract | Yes |
| Introduction includes at least three parts: background, related work, and motivation | Yes |
| A pipeline/network figure is provided | Figure. 1 |
| Pre-processing | Page 4 |
| Strategies to data augmentation | Page 4 |
| Strategies to improve model inference | None |
| Post-processing | None |
| Environment setting table is provided | Table 1 |
| Training protocol table is provided | Table 2 |
| Ablation study | Page 6 |
| Efficiency evaluation results are provided | None |
| Visualized segmentation example is provided | Figure 2 |
| Limitation and future work are presented | Yes |
| Reference format is consistent. | Yes |
| Main text $>= 8$ pages (not include references and appendix) | Yes |