# OpenReview forum: "Optimizing SAT-Nano for Text-Guided 3D Medical Segmentation using Dice Semimetric Loss"
_thecvf.com/CVPR/2025/Workshop/MedSegFM — CVPR 2025 Workshop MedSegFM Submission_

### Official Review · Reviewer_LR6b · 2025-09-16
**Review for Optimizing SAT-Nano for Text-Guided 3D Medical Segmentation using Dice Semimetric Loss**

**Rating:** 3
**Confidence:** 5

**Review:**

Summary:
This paper proposes SAT-Nano-JDT, which integrates JDTLoss with standard Dice and BCE losses under deep supervision for text-guided 3D medical segmentation. The method is evaluated on the CVPR MedSegFM Challenge validation dataset across four imaging modalities.
Results show modest improvements for,
CT segmentation (DSC: 0.604 → 0.644) and,
Limited improvements for MRI (0.448 → 0.426),
while demonstrating significant performance degradation on PET (0.168 → 0.029) and mixed results on microscopy (0.554 → 0.292).

Strengths:
1. Clear positioning within established challenge framework: The work is appropriately situated within the CVPR MedSegFM challenge context.
2. Mathematically sound formulation: The JDTLoss integration is correctly formulated with proper mathematical notation and citations.
3. Comprehensive experimental documentation: Training parameters and environment details are thoroughly documented in tables.
4. Transparency in reporting: The authors honestly report negative and mixed results rather than overselling their contribution.
5. Code availability: A repository is provided for reproducibility.

Weaknesses: Scientific Integrity Concerns
- The checklist in Table 4 claims "Ablation study: Yes (Page 6)." However, page 6 contains only Table 3, which presents a baseline-versus-proposed comparison. This is not an ablation study and represents a factual misstatement in the submission.

Experimental Limitations

1. Lack of proper ablation analysis: No systematic study of individual loss components (Dice, BCE, JDT) or their relative contributions.
2. Missing comparative analysis: The paper references both DML1 and DML2 variants but provides no comparison between them.
3. Unjustified design choices: Equal weighting of loss components is adopted without empirical justification or sensitivity analysis.
4. Insufficient statistical validation: Results appear to be single-run experiments without error bars or significance testing.
5. Missing training dynamics: No convergence curves or learning behavior analysis is provided.

Empirical Concerns

1. Marginal improvements: CT improvements of +0.001 DSC are within measurement noise.
2. Severe performance degradation: PET F1 score drops catastrophically from 0.113 to 0.012.
3. Inconsistent modality performance: The method improves some modalities while significantly degrading others, without adequate explanation.

Technical and Methodological Issues

1. Insufficient technical detail: Deep supervision implementation is mentioned but not described (which layers, what weighting scheme).
2. Unspecified variants: "DML variant" is referenced without clarification.
3. Missing computational analysis: No discussion of computational overhead or efficiency considerations.
4. Inadequate failure analysis: Section 4.3 is incomplete, and the explanation of modality-specific failures is superficial.
5. Limited discussion of text-prompt influence: The role of text guidance in the observed performance patterns is not analyzed.

Recommendation: Reject
While the core idea of integrating JDTLoss with SAT-Nano has merit, the paper suffers from significant methodological and integrity issues. The false claim regarding ablation studies is particularly concerning. The empirical contribution is limited, with improvements that are either marginal or accompanied by substantial degradations in other modalities.
To be suitable for publication, the authors should:

1. Provide genuine ablation studies analyzing individual loss components
2. Include proper statistical validation with multiple runs and significance testing
3. Conduct a thorough failure analysis explaining modality-specific behavior patterns
4. Remove false claims from the checklist and ensure factual accuracy
5. Complete the missing analysis sections
6. Justify design choices through empirical evidence

The work would benefit from a more systematic experimental approach and a deeper analysis of the observed phenomena before resubmission.

Rating: 3/10 (Clear rejection)
Confidence: 5/5 (Confident in evaluation)

---

### Official Review · Reviewer_jAYF · 2025-09-16
**Replacement of Loss Function Lacks Substantive Innovation**

**Rating:** 4
**Confidence:** 5

**Review:**

This paper proposes SAT-Nano-JDT, a variant of the SAT-Nano baseline model for text-guided 3D medical image segmentation. The main contribution is the integration of JDTLoss (Dice Semimetric Loss) into the loss function to directly optimize the Dice score, aiming to improve segmentation accuracy. The authors fine-tune the SAT-Nano model on the CVPR 2025 challenge coreset (10% of data). Experimental results show marginal improvements over the baseline in some cases, such as CT semantic segmentation (DSC 0.644 vs. 0.643 baseline) and microscopy instance DSC TP (0.310 vs. 0.292 baseline). However, performance on other modalities such as MRI, PET, and Ultrasound is worse than or equal to the baseline. The authors conclude that while JDTLoss is theoretically sound, its integration provides only limited empirical benefits.

**Strengths:**
1. The paper addresses a relevant and timely challenge in text-guided 3D medical image segmentation, which is an emerging area of interest.
2. The proposed use of *JDTLoss* is conceptually straightforward and well motivated, as it directly optimizes the Dice score, a key evaluation metric in segmentation tasks.
3. The methodology is clearly described, with sufficient details on architecture, dataset, and training protocols to enable reproducibility.
4. The paper is well organized and clearly written, making it easy to follow the motivation, approach, and results.
5. Limitations are acknowledged, and the authors are transparent about the modest improvements and failure cases.

**Weaknesses:**
1. The contribution is essentially the substitution of the loss function in an existing strong baseline (SAT-Nano). There are no architectural innovations or new methodological insights beyond loss integration. And even the JDLoss, is a published work on MICCAI 2023.
2. Results show negligible or inconsistent gains (e.g., CT DSC improves by 0.001, microscopy improves slightly), while several modalities degrade significantly (e.g., PET F1 drops from 0.113 to 0.012). This undermines the claim of effectiveness.
3. The effect of JDTLoss relative to Dice loss or BCE is not isolated. The weighting strategy for the composite loss is fixed without justification or tuning.
4. The results do not demonstrate meaningful improvements over the baseline, making the practical and scientific impact of the work limited.

---

### Official Review · Reviewer_JGxB · 2025-09-27
**Limited Impact of JDTLoss on the SAT model Baseline**

**Rating:** 4
**Confidence:** 5

**Review:**

This paper introduces SAT-Nano-JDT, an adaptation of the SAT-Nano baseline fine-tuned by integrating JDTLoss (Dice Semimetric Loss) to directly optimize the Dice score for 3D biomedical segmentation. The empirical results on the coreset validation set are mixed and marginal: while incremental gains were noted for CT and Microscopy, the model failed to consistently surpass the strong SAT-Nano baseline across all modalities, performing equal to or worse on MRI and PET. This suggests the isolated loss modification provided limited practical value.

Strengths:
1. The use of JDTLoss is conceptually sound and directly targets the optimization of the Dice score, the primary evaluation metric.
2.Clarity and Transparency: The methodology is clearly described. The authors are commendable for their transparency regarding the modest empirical results and the acknowledged limitations of their approach.

Weaknesses:
1. The reported improvements are inconsistent and too small to demonstrate a robust enhancement over the strong baseline. The model's effectiveness is not clearly proven.
2. The theoretical benefits of JDTLoss are not realized in practice; its integration yields insufficient practical improvement, undermining the core contribution.
3. The paper lacks a deep analysis explaining why JDTLoss failed to improve challenging cases. The authors must more thoroughly analyze the task's actual difficulties and explicitly link the JDTLoss's properties to the failure modes.

---

### Decision · Program_Chairs · 2025-11-12

Revision